# Empagliflozin Prevent High-Glucose Stimulation Inducing Apoptosis and Mitochondria Fragmentation in H9C2 Cells through the Calcium-Dependent Activation Extracellular Signal-Regulated Kinase 1/2 Pathway

**DOI:** 10.3390/ijms25158235

**Published:** 2024-07-28

**Authors:** Yung-Lung Chen, Hui-Ting Wang, Wen-Chin Lee, Pei-Ting Lin, Wen-Hao Liu, Shu-Kai Hsueh

**Affiliations:** 1Division of Cardiology, Department of Internal Medicine, Kaohsiung Chang Gung Memorial Hospital, Chang Gung University College of Medicine, Kaohsiung 833, Taiwan; r40391132@gmail.com (P.-T.L.); wenhao@cgmh.org.tw (W.-H.L.); pather@cgmh.org.tw (S.-K.H.); 2Graduate Institute of Clinical Medical Sciences, College of Medicine, Chang Gung University, Taoyuan 333, Taiwan; 3School of Medicine, College of Medicine, National Sun Yat-sen University, Kaohsiung 804, Taiwan; gardinea1983@gmail.com; 4Emergency Department, Kaohsiung Chang Gung Memorial Hospital, Kaohsiung 833, Taiwan; 5Division of Nephrology, Department of Internal Medicine, Kaohsiung Chang Gung Memorial Hospital, Chang Gung University College of Medicine, Kaohsiung 833, Taiwan; leewenchin@gmail.com

**Keywords:** calcium-dependent, empagliflozin, extracellular signal-regulated kinase 1/2, H9C2, high-glucose stimulation, mitochondria fragmentation

## Abstract

A previous study showed that high-glucose (HG) conditions induce mitochondria fragmentation through the calcium-mediated activation of extracellular signal-regulated kinase 1/2 (ERK 1/2) in H9C2 cells. This study tested whether empagliflozin could prevent HG-induced mitochondria fragmentation through this pathway. We found that exposing H9C2 cells to an HG concentration decreased cell viability and increased cell apoptosis and caspase-3. Empagliflozin could reverse the apoptosis effect of HG stimulation on H9C2 cells. In addition, the HG condition caused mitochondria fragmentation, which was reduced by empagliflozin. The expression of mitochondria fission protein was upregulated, and fusion proteins were downregulated under HG stimulation. The expression of fission proteins was decreased under empagliflozin treatment. Increased calcium accumulation was observed under the HG condition, which was decreased by empagliflozin. The increased expression of ERK 1/2 under HG stimulation was also reversed by empagliflozin. Our study shows that empagliflozin could reverse the HG condition, causing a calcium-dependent activation of the ERK 1/2 pathway, which caused mitochondria fragmentation in H9C2 cells.

## 1. Introduction

Type 2 diabetes mellitus poses a significant risk to human well-being, leading to various consequences affecting small blood vessels (microvascular) and large blood vessels (macrovascular), which include retinopathy, nephropathy, cardiovascular disease, and peripheral arterial disease [1]. It is estimated that there are just under half a billion people living with diabetes around the globe, and it is anticipated that this figure will increase by 25% in the year 2030 and by 51% in the year 2045. at the very least [2]. Studies have demonstrated that sodium–glucose cotransporter 2 inhibitor (SGLT2i) benefits the heart and kidneys in individuals with diabetes [3,4]. The cardiorenal protective benefits of SGLT2 inhibitors have been widely recognized due to positive outcomes observed in major clinical trials such as EMPA-REG OUTCOME, DAPA-HF, DAPA-CKD, CANVAS, and CREDENCE [3,4,5,6,7]. It is important to note that clinical guidelines recommend SGLT2 inhibitors as the glucose-lowering agent for patients with diabetes who have a higher cardiovascular risk or renal insufficiency [8,9].

A number of hypotheses have been put forward to explain the renal and cardiovascular advantages associated with SGLT2 inhibitors. Within the proximal tubules, it has been demonstrated that SGLT2 inhibitors can directly lower the oxidative stress that is caused by elevated glucose levels [10,11]. Additional pathways include a decrease in glomerular hyperfiltration through tubuloglomerular feedback, a reduction in inflammation, an amelioration of fibrosis, a reduction in the stimulation of the sympathetic nervous system, and an improvement in mitochondrial function [10,11,12,13,14]. Curiously, the expression of SGLT2 is absent in the adult heart. The cardioprotective effects of SGLT2 inhibitors are believed to be caused by several mechanisms. These include osmotic diuresis, which helps remove excess fluid from the body; natriuresis, which promotes the excretion of sodium; improved renal function, which enhances the kidneys’ ability to filter waste products, provides a reduction in blood pressure, an improvement in vascular function, an increase in hematocrit levels, changes in how the body handles sodium in tissues, a decrease in inflammation caused by adipose tissue, a decrease in the production of proinflammatory cytokines, a shift towards using ketone bodies as the main source of energy, a decrease in serum uric acid levels, the suppression of advanced glycation end product signaling, and a decrease in oxidative stress [15,16]. The majority of these mechanisms consider the regulatory interaction between the kidney and heart.

Previous studies showed that high-glucose (HG) conditions induce mitochondria fragmentation through the calcium-mediated activation of the extracellular signal-regulated kinase 1/2 (ERK 1/2) pathway in the H9C2 cell line [17]. An earlier study discovered that empagliflozin safeguards HK-2 cells from HG-mediated harm by means of a mitochondrial mechanism [10]. It also mitigates diabetic tubulopathy by reducing mitochondrial fragmentation and autophagy in human renal proximal tubular cells (hRPTCs) and streptozotocin-induced diabetic rodents [18]. Empagliflozin has also been demonstrated to improve the function of mitochondria and reduce oxidative stress in diabetic cardiomyopathy in a previous study [19]. Empagliflozin reduces the profibrotic activities of atrial fibroblasts by inhibiting a Na+/H+ exchange (NHE) [20]. The discharge of endoplasmic reticulum Ca^2+^ and the entrance of extracellular Ca^2+^ are both diminished [20]. Calcium activation of ERK has been reported through a variety of pathways, such as PYK2, EGF receptor transactivation, RasGRF1, and CalDAG-GEF and CaM kinases [21]. Previous research demonstrated that the effects of empagliflozin on ERK1/2 vary depending on the type of cellular model. Empa suppresses the activity of ERK1/2 in models of hepatocellular carcinoma, renal tubules, and microglia [22,23,24]. Nevertheless, both the TAC and Myocardial I/R models demonstrated that empagliflozin has the ability to restore p-ERK1/2 and, hence, achieve the cardioprotective effect [25,26].

It was unknown whether empagliflozin could prevent HG-induced cellular apoptosis and mitochondria fragmentation in H9C2 cells by inhibiting the calcium-mediated ERK1/2 pathway. The objective of this investigation was to determine whether empagliflozin could prevent HG-induced cellular apoptosis and mitochondria fragmentation in the H9C2 cell line by inhibiting the calcium-mediated activation of the ERK1/2 pathway.

## 2. Results

### 2.1. Empagliflozin Ameliorates the HG-Caused Decrease in H9C2 Viability

After a treatment period of at least 24 h, the results of the test that examined the dose-dependent and time-dependent effects of glucose revealed that the cell viability experienced a substantial decrease under the doses of 25, 35, and 45 mM (all *p* < 0.05) (Appendix A). For further analysis, the HG condition was selected to be treated with a dose of 35 mM for a period of twenty-four hours. In order to evaluate the impact of higher glucose (HG) circumstances and empagliflozin on the viability of H9C2 cells, we conducted experiments to determine the impact of empagliflozin at doses of 0.5, 1, and 5 μM on the HG conditions. As demonstrated in Figure 1, the presence of the HG condition resulted in a 10% decrease in the viability of H9C2 cells when compared to the regular glucose condition (*p* < 0.01). Despite the fact that empagliflozin did not have any effect on cellular viability, the reduction in cell viability that was brought about by the HG condition was alleviated by the administration of 1 μM empagliflozin when it was treated (*p* < 0.05).

### 2.2. Empagliflozin Rescues HG-Induced Apoptosis in H9C2 Cells

For the purpose of determining the impact that empagliflozin has on the apoptosis that is generated by HG in H9C2 cells, we examined the TUNEL labeling and the expression of caspase-3 in H9C2 cells that had been treated with both HG and empagliflozin. In Figure 2A,B, we have an illustration of the quantitative analysis of cellular apoptosis in H9C2 cells as well as the positive TUNEL staining in each of the four treatment groups. An examination of the expression of caspase-3 was carried out using Western blotting, and the quantitative analysis is depicted in Figure 2C,D. Apoptosis was significantly induced by HG, and empagliflozin, when administered at a dose of either 1 or 5 μM, was able to rescue the apoptosis that was generated by the HG condition (all *p* < 0.05).

### 2.3. Empagliflozin Improves the Mitochondrial Fragmentation of HG-Treated H9C2 Cells

In order to determine the impact that HG and empagliflozin have on the fragmentation of mitochondria in H9C2 cells, we examined the mitochondrial fragmentation that occurred in H9C2 cells that had been treated with HG and empagliflozin. Figure 3A–E depicts the morphological properties of mitochondria. These characteristics were obtained from the control group as well as the four treatment groups. Figure 3F–J presents magnified pictures of mitochondria that have been indexed among the various treatment groups. We found that H9C2 cells were susceptible to mitochondrial fragmentation that was produced by HG and that empagliflozin was able to reduce the severity of this event. We calculated the rate of mitochondrial fission in each treatment group so that we could conduct a quantitative analysis of the process of mitochondrial fragmentation. Under normal glucose conditions, the fission rate in H9C2 cells was sixty times lower than in HG conditions. This was the effect of the normal glucose condition. When HG is present, the mitochondrial fission rate can be decreased to a level that is comparable to what is found under normal glucose settings. This can be accomplished by administering empagliflozin at either 1 or 5 μM doses, as shown in Figure 3K.

### 2.4. Empagliflozin Reduces the Expression Levels of Mitochondrial Fission Proteins of H9C2 Cells Incubated in HG Condition

For the purpose of determining the impact that empagliflozin has on the levels of expression of mitochondrial fission and fusion proteins, we carried out Western blotting investigations of DRP1, FIS1, MFN1, and MFN2 in H9C2 cells that were subjected to four different treatment groups (Figure 4A). According to the data presented in Figure 4B,C, the expression of the DRP1 and FIS1 proteins increased when the HG condition was present. Under the administration of empagliflozin, however, the same proteins were reduced to a level comparable to that found in the normal glucose state, particularly at doses of 1 or 5 μM. At the same time, the expression of the fusion proteins MFN1 and MFN2 was increased in the HG condition; however, when the glucose levels were normalized with empagliflozin, there was no statistically significant difference between the HG condition and normal glucose compared to the fusion proteins.

### 2.5. Empagliflozin Reduces HG-Induced Intracellular Calcium Overload Accumulation in H9C2 Cells

By analyzing the intracellular Ca^2+^ ([Ca^2+^]i) with Fluo-4 AM fluorescence in four different treatment groups, we were able to determine the effect that empagliflozin had on the accumulation of the intracellular calcium overload that was generated by HG (Figure 5A). HG therapy resulted in an increase in [Ca^2+^]i, while empagliflozin was able to prevent the formation of intracellular calcium excess. This is demonstrated in Figure 5B.

### 2.6. Impact of Empagliflozin on the ERK1/2 Activation, Which Leads to Mitochondrial Fragmentation in HG-Treated H9C2 Cells

To determine the effect that empagliflozin has on the ERK1/2-induced mitochondrial fragmentation that occurs in H9C2 cells treated with HG, we examined the activation of ERK1/2 in four different treatment groups. As demonstrated in Figure 6, empagliflozin was able to reduce the amount of p-ERK1/2 expression that was elevated as a result of the HG therapy.

## 3. Discussion

In this investigation, we discovered that empagliflozin effectively prevented apoptosis of the H9C2 cell line produced by high glucose (HG). It achieved this by reducing mitochondrial fragmentation, mostly by suppressing the activity of mitochondria fission proteins (FIS-1 and DRP-1) through the inhibition of the calcium-dependent activation of the ERK1/2 pathway.

Previous research has shown that the treatment of H9C2 cells with HG leads to a substantial reduction in cell viability as a consequence of cellular apoptosis and mitochondrial fragmentation [27,28]. Mitochondrial fission is inhibited, and apoptosis induced by HG is prevented in retinal endothelial cells and renal tubular cells through the downregulation of DRP1 and FIS1 [10,29]. It has been discovered that empagliflozin, a novel oral antidiabetic drug that is an SGLT2i, possesses cardioprotective and renoprotective properties in patients who are diabetic as well as people who do not have diabetes [3,4,30]. Empagliflozin ameliorates diabetic cardiomyopathy by reducing oxidative stress and enhancing mitochondrial function, as demonstrated in a previous study [19]. In palmitate-treated H9C2 cells, empagliflozin significantly increased the expression of mitochondrial fusion-related proteins (MFN1 and OPA1, but not MFN2) and inhibited the expression of DRP1. These results suggest that empagliflozin may enhance mitochondrial function by inhibiting mitochondrial fission in the heart of diabetes. Nevertheless, the research did not assess the expression of FIS1. Empagliflozin inhibits mitochondrial fission during ischemia/reperfusion injury and normalizes the size and number of mitochondria through the predominant mechanism of FIS1 dephosphorylation, as demonstrated by other studies [31,32]. Our findings indicated that empagliflozin safeguarded H9C2 from HG-induced injuries and mitochondrial fragmentation by reducing the production of caspase-3 and modulating the expression of mitochondrial fusion and fission proteins. When compared to the protein level, the HG condition resulted in a considerable increase in the expression levels of both mitochondrial fission proteins (DRP1 and FIS1). On the other hand, the administration of empagliflozin medication was able to reverse this impact by decreasing the levels of expression of both mitochondrial fission proteins. The expression levels of proteins that are responsible for mitochondrial fusion appeared to increase in response to elevated glucose levels. Conversely, empagliflozin normalized the number of these proteins in the HG state. Through the regulation of the expression of fusion proteins and the reduction of the quantities of fission proteins, empagliflozin may improve the mitochondria’s ability to withstand fragmentation that is generated by HG levels. This is in accordance with the protective mechanism that has been theorized in a variety of cellular activities [10,19,33]. Dysregulated mitochondrial homeostasis induces cardiac dysfunction and remodeling, which, in turn, leads to cardiovascular disease and complications due to the high energy demands of cardiomyocytes as they circulate blood throughout the body under typical physiological conditions [34]. The post-translational modification of mitochondrial proteins, mitochondrial dynamics (biogenesis, fission, and fusion), and mitochondrial autophagy are all components of mitochondrial quality control [35]. The accumulation of evidence indicates that mitochondrial quality control in cardiomyocytes has the potential to enhance cardiac function, rescue failing cardiomyocytes, and prevent the progression of cardiovascular disease in response to external environmental stress [36]. This may account for the favorable effect of empagliflozin on cardioprotection in clinical trials.

There is a possibility that empagliflozin’s ability to prevent HG-induced mitochondrial fragmentation and cellular apoptosis in H9C2 cells can be explained by some protective mechanisms. According to the findings of a previous study, the osmotic effect of glucose causes the activation of the G protein(s) through a stretch receptor. This activation then leads to the stimulation of calcium channels, which ultimately results in the influx of calcium into the cardiac myocytes [37]. The activity of NHE is diminished by empagliflozin, resulting in a reduction in the production of phosphorylated phospholipase c and inositol 1,4,5-triphosphate [20]. Accordingly, this results in a reduction in the release of calcium from the endoplasmic reticulum as well as the entry of calcium from extracellular sources into the atrial fibroblasts [20]. In the course of our research, we found that the levels of Ca^2+^ within the cells increased as a result of the HG treatment and that empagliflozin was able to reduce the accumulation of calcium overload within the cells. Prior research has also demonstrated that empagliflozin has a considerable impact on the control of calcium ions, late sodium currents, and NHE currents, as well as the electrophysiological characteristics of diabetic cardiomyopathy [38]. These effects may play a role in the cardioprotective benefits of empagliflozin in individuals with diabetes mellitus. Furthermore, findings from a previous study indicated that ERK1/2 was activated by HG incubation in a manner that was dependent on Ca^2+^ [17]. The Ca^2+^ chelation prevented the activation of ERK1/2 and also inhibited the phosphorylation of ERK1/2, which led to a reduction in the amount of mitochondrial fragmentation that occurred in response to HG stimulation. Furthermore, in vitro kinase experiments indicated that ERK1/2 is capable of phosphorylating DRP1 in a manner that was demonstrated. Through the use of empagliflozin, we were able to demonstrate that it reduced the concentration of Ca^2+^ within the cell, which, in turn, prevented the activation of ERK1/2 by blocking the phosphorylation of ERK1/2. A previous in vivo study demonstrated that short-term empagliflozin treatment was associated with a significant downregulation of insulin receptor protein expression in db/db mice [39]. Additionally, empagliflozin treatment did not alter the phosphorylation of ERK1/2. Nevertheless, a separate investigation demonstrated that empagliflozin decreases the expression of pro- and anti-inflammatory mediators [23]. This effect may be mediated by NHE-1 and the subsequent inhibition of the ERK1/2 and NFkB pathways. In vitro and in vivo studies should be conducted to further assess these distinctions. Additionally, this resulted in a decrease in the levels of fission proteins (DRP1 and FIS1) in response to HG stimulation, which led to a reduction in mitochondrial fragmentation. The EMPA-REG OUTCOME trial’s demonstration of empagliflozin’s cardioprotective effects could be ascribed to a unique mechanism discovered throughout the investigation. As a result, additional research is required to ascertain its role in diabetic cardiomyopathy.

This study had several limitations. First, this investigation exclusively assessed the in vitro impact of the empagliflozin treatment on the HG condition. The results indicated that empagliflozin’s protective effects are mediated through the calcium-dependent activation of the extracellular signal-regulated kinase 1/2 pathway. In order to verify this mechanism, additional in vivo research should be conducted. Secondly, the impact of siRNA knockdown of ERK1/2 was not evaluated in this study. Further in vitro and in vivo research utilizing siRNA knockdown of ERK1/2 to assess its impact on HG-induced mitochondria fragmentation would be beneficial in enhancing the protective effects of empagliflozin through this pathway.

In conclusion, the results of our research indicate that the HG condition, which is characterized by the calcium-dependent activation of the ERK 1/2 pathway, can be reversed through the administration of empagliflozin therapeutics. This particular mechanism is accountable for the fragmentation of mitochondria in H9C2 cells’ mitochondria. The treatment with empagliflozin improves mitochondrial function in H9C2 cells, which helps to reduce the negative consequences of lesions caused by HG.

## 4. Materials and Methods

### 4.1. Cell Culture and Chemicals

H9C2 cardiomyoblast cells (ATCC CRL-1446, ECACC 88092904) were maintained in low glucose Dulbecco’s modified Eagle’s medium (Gibco; Thermo Fisher Scientific, Inc.,Waltham, MA, USA) supplemented with 10% fetal bovine serum (Gibco; Thermo Fisher Scientific, Inc.,Waltham, MA, USA), 1% penicillin-streptomycin solution (Gibco) at 37 °C, 5% CO_2_, and 95% humidity. Cells were incubated with normal (5 mM) or high (25, 35, and 45 mM) D-glucose (Gibco; Thermo Fisher Scientific, Inc.,Waltham, MA, USA) for 24, 48, 72 h. In order to evaluate the protective impact of empagliflozin (EMPA; Cayman Chemical, Inc., Ann Arbor, MI, USA) on the HG condition, H9C2 cells were subjected to high glucose or in conjunction with various concentrations of empagliflozin, with the final concentration being set at 0.5, 1 and 5 µM, respectively.

### 4.2. Cell Viability Assay

For the purpose of determining the viability of the cells, the PrestoBlue assay (Invitrogen; Thermo Fisher Scientific, Inc.,Waltham, MA, USA) was utilized in accordance with the directions provided by the manufacturer. Further, 96-well plates were used to culture a total of 3 × 10^3^ cells per well, and the cells were treated in the same manner as described earlier [40]. The treatment was followed by the addition of 10 μL of PrestoBlue solution to every well. The cells were then incubated for a further three hours. Following this, the absorbance was measured at wavelengths of 570 nm for excitation and 600 nm for emission.

### 4.3. Western Blot Analysis

A plate with six wells was used to distribute cells at a density of 3 × 10^5^ cells per well. After overnight incubation, the cells were placed in either a normal medium or an HG media for further incubation. The examination of the Western blot was carried out in accordance with the methods that were stated earlier [41]. Concisely, the cell pellets were disrupted in a buffer and then centrifuged at a force of 14,000× *g*. A total of 20 μg of protein, found in the supernatant of every sample, was separated by sodium dodecyl sulfate polyacrylamide gel electrophoresis (SDS-PAGE). Subsequently, the protein was transferred onto polyvinylidene difluoride membranes through the process of electrophoresis. The membranes were submerged in TBST buffer and kept from undergoing any further reactions for a period of one hour while the temperature remained at room temperature. Primary antibodies against MFN1 (sc-100561, Santa Cruz Biotechnology, Inc., Dallas, TX, USA), MFN2 (sc-100560, Santa Cruz Biotechnology, Inc.), DRP1 (sc-101270, Santa Cruz Biotechnology, Inc., Dallas, TX, USA), FIS1 (GTX111010, GeneTex, Inc., Irvine, CA, USA), caspase-3 (9664S, Cell signaling, Inc., Danvers, MA, USA), and ERK 1/2 (05-1152, Millipore, Inc., Burlington, MA, USA) were utilized in the analysis of the blot-based samples. After that, the blots were subjected to incubation by secondary antibodies that were diluted to a suitable concentration. The Western Lightning Plus-ECL (PerkinElmer, Inc., Waltham, MA, USA) was utilized to observe the blots.

### 4.4. Mitochondrial Fragmentation Index

The mitochondrial morphology was evaluated in accordance with the methods that were given earlier [10], with a few alterations brought about by modifications. The mitochondria were stained with the fluorescent dye Mitotracker red (Invitrogen, Life Technologies, Inc., Carlsbad, CA, USA), and then they were inspected with a confocal microscope (LSM 700, CarlZeiss microscopy, Inc., White Plains, NY, USA). In summary, the mitochondria were stained. The shape of mitochondria was examined separately and without any outside influence by two qualified technicians. Those mitochondria that had punctuated or fragmented patterns were considered to be of the fission type. A calculation was made to determine the fission rate (%) by multiplying the ratio of the fission type to the total number of mitochondria (fission, intermediate, and fusion) by 100, respectively.

### 4.5. Cellular Apoptotic Assay

In the terminal deoxynucleotidyl transferase dUTP nick end labeling (TUNEL) test, H9C2 cells were fixed in 4% paraformaldehyde and then treated with a blocking solution consisting of 10% FBS in PBS for a duration of 30 min at a temperature of 25 °C. Whether empagliflozin was present or absent, the cells were allowed to incubate for a period of 24 h. Immediately following the permeabilization of the cells with 0.1% (*v*/*v*) Triton X-100 in PBS, the TUNEL reaction mixture (Roche, Mannheim, Germany) was then incubated with the cells for one hour at 37 °C. The cells were examined under a fluorescence microscope (Olympus BX-51; Olympus Corporation, Tokyo, Japan) following the treatment.

### 4.6. Calcium Live Cell Imaging

The calcium indicator dye, Fluo-4 AM (Invitrogen, Life Technologies, Inc., Carlsbad, CA, USA), was employed to ascertain the intracellular Ca^2+^ overload. In summary, the cells from all groups were incubated with Fluo-4 AM/power load/Probenecid in live cell imaging solution for 30 min at 37 °C, followed by two washes with PBS. The cells were subsequently observed using a fluorescence microscope (Olympus IX-51; Olympus Corporation, Tokyo, Japan) at an excitation/emission wavelength of 494/506 nm. ImageJ software (U.S. National Institutes of Health, Bethesda, MD, USA) was employed to analyze the images.

### 4.7. Statistical Analysis

Each experiment was conducted at least three times. The quantitative data are represented as the mean ± standard error of the mean (SEM). IBM SPSS Statistics for Windows, Version 19.0 (IBM Corp., Armonk, New York, NY, USA) was employed to conduct statistical analyses. One-way analysis of variance (ANOVA) was employed to conduct comparisons between multiple categories. The *t*-test was employed to conduct pair-wise comparisons. A value of *p* < 0.05 was regarded as statistically significant.

## Figures and Tables

**Figure 1 ijms-25-08235-f001:**
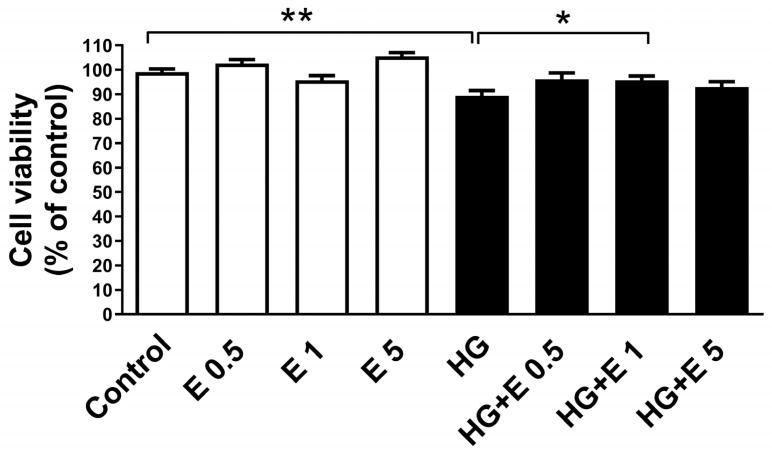
Effect of high-glucose condition and empagliflozin on H9C2 cells in cell viability. Cell viability was measured with PrestoBlue reagent assay and the absorbance was measured at wavelengths 570 nm, using 600 nm as a reference wavelength. Results are mean ± SEM (N = 7). Only high glucose (35 mM) treatment led to a decrease in cell viability. * *p* < 0.05; ** *p* < 0.01 vs. the control group (5 mM) of 24 h. HG, high glucose; E, empagliflozin. (N = 7).

**Figure 2 ijms-25-08235-f002:**
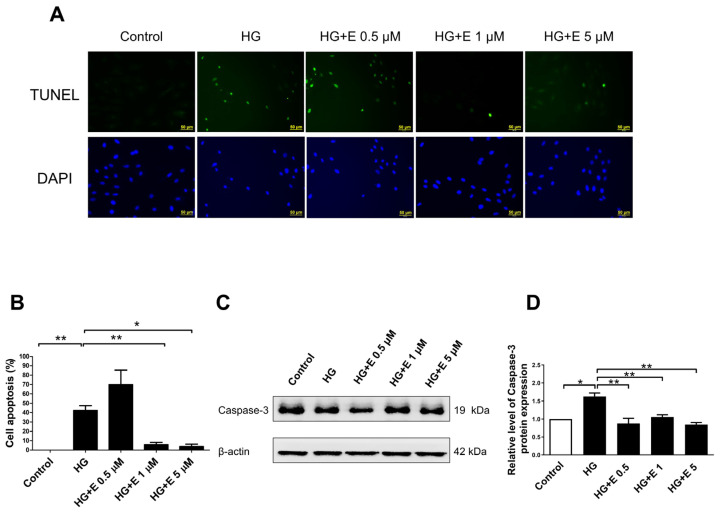
Effects of empagliflozin on high glucose-induced apoptosis in H9C2 cells. (**A**) Fluorescence pictures demonstrate the presence of positive TUNEL staining in all four treatment groups. (**B**) Quantitative analysis of the TUNEL assay. (**C**,**D**) The level of caspase-3 expression was assessed using Western blotting and then standardized based on the level of β-actin. The study demonstrates that elevated glucose levels significantly trigger apoptosis, but empagliflozin mitigates this impact. The data are presented as the mean value ± the standard error of the mean (SEM). * *p* < 0.05; ** *p* < 0.01 vs. the control group (5 mM) of 24 h. HG, high glucose; E, empagliflozin. (TUNEL stain, N = 4, Western Blot: Caspase-3, N = 3).

**Figure 3 ijms-25-08235-f003:**
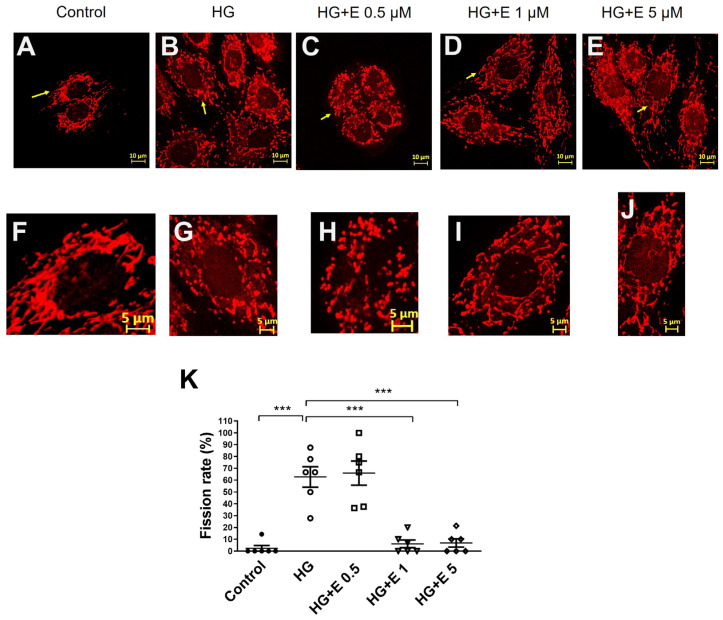
Empagliflozin improves the mitochondrial fragmentation of high glucose-treated cardiomyocytes. (**A**,**B**) Compared with Glu 35 mM, Mitochondria fragmentation increased in H9C2 cells during high glucose injury. Scale bar: 10 μm. (**C**–**E**) Empagliflozin decreased the quantity of fragmentation mitochondria in H9C2 cells during high glucose injury. (**F**–**J**) The image showed individual cells in different conditions. Scale bar: 5 μm. (**K**) The mitochondrial fission rate was higher in high glucose condition. Empagliflozin improved this effect. Data are expressed as mean ± SEM. *** *p* < 0.001 vs. the control group (5 mM) of 24 h. (Mitotracker red, N = 6).

**Figure 4 ijms-25-08235-f004:**
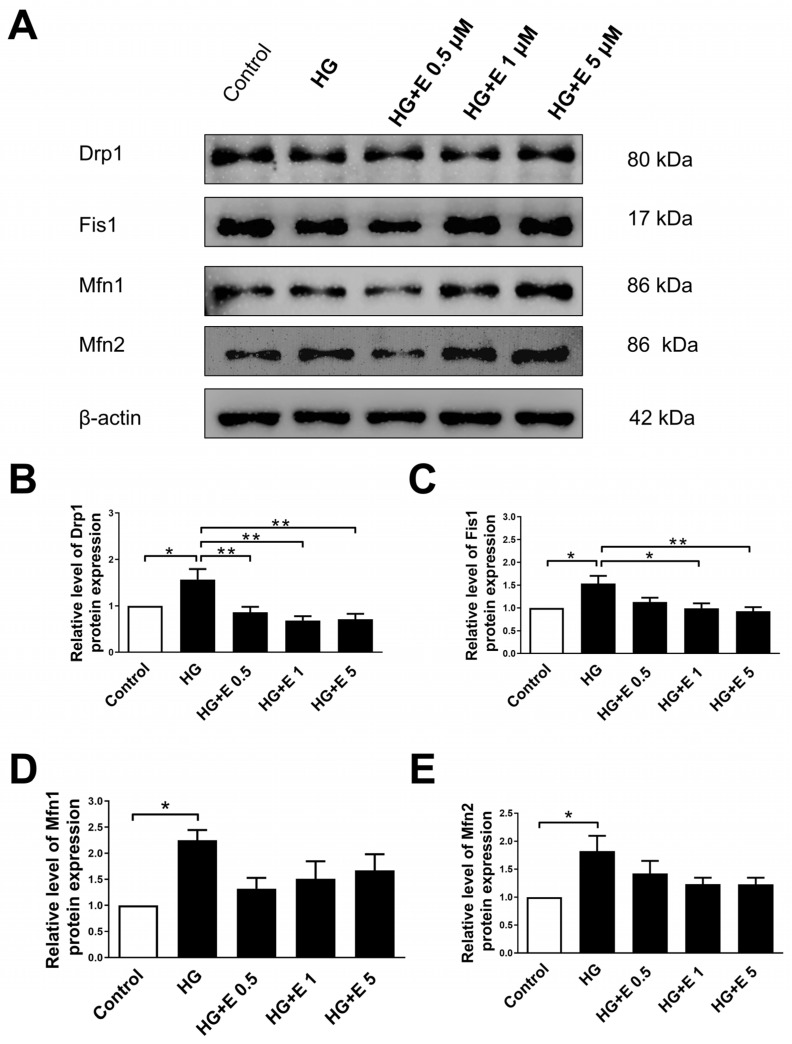
The effect of empagliflozin on the changes in the levels of mitochondrial fusion/fission proteins generated by high glucose. (**A**–**E**) The expression of DRP1, FIS1, MFN1, and MFN2 were analyzed with Western blotting and normalized to the level of β-actin. Data are expressed as mean ± SEM. * *p* < 0.05; ** *p* < 0.01 vs. the control group (5 mM) of 24 h. (Western blot: Drp1, N = 5; Fis1, N = 4; Mfn1, N = 4; Mfn2, N = 5).

**Figure 5 ijms-25-08235-f005:**
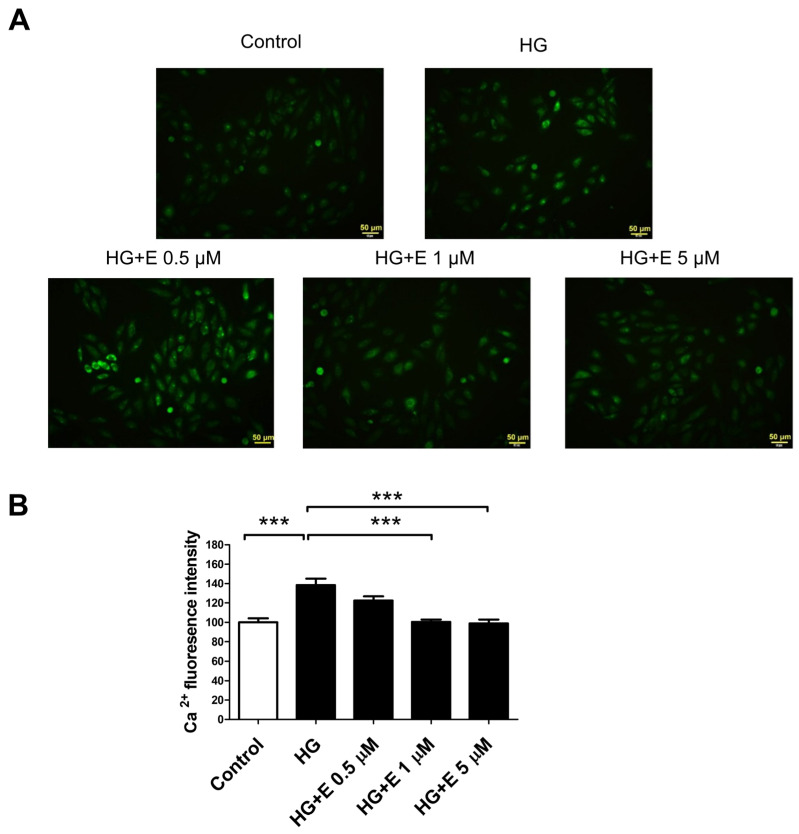
Effects of empagliflozin on HG-induced intracellular calcium overload accumulation in H9C2 cells. (**A**) Fluo-4 AM fluorescence showing the [Ca^2+^]i. Scale bar = 50 μm and (**B**) Relative fluorescent intensity. Values are mean ± SEM. *** *p* < 0.001 vs. the control group (5 mM) of 24 h. (Fluo 4, N = 4).

**Figure 6 ijms-25-08235-f006:**
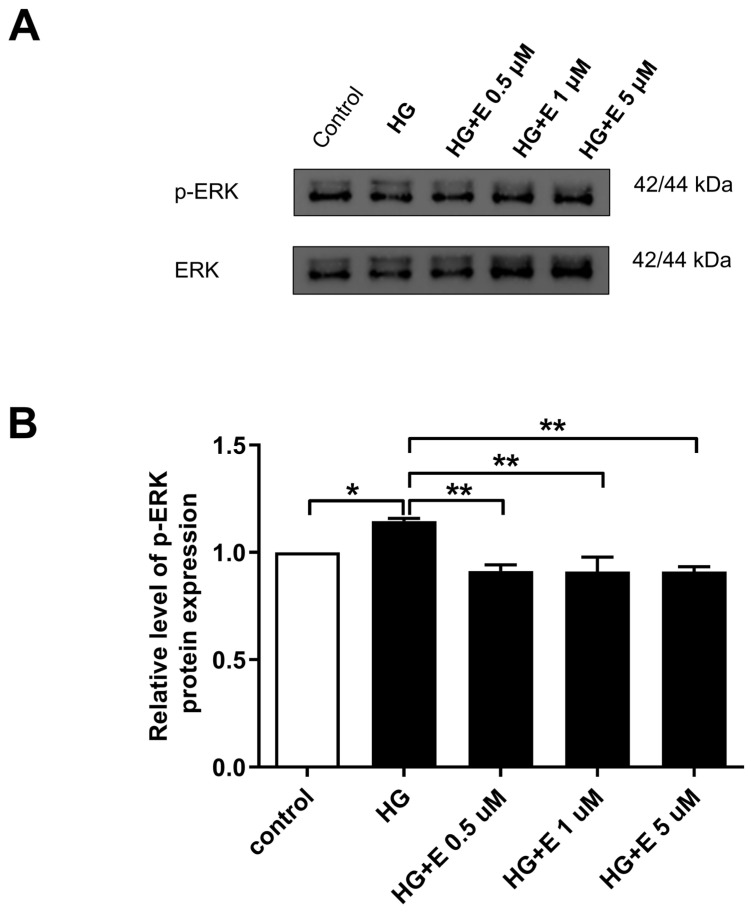
Empagliflozin alleviated ERK1/2 activation leading to mitochondrial fragmentation of HG-treated H9C2 cells. (**A**,**B**) The expression of ERK1/2 was analyzed with Western blotting and normalized to the level of ERK. Data were obtained from three independent experiments and are expressed as mean ± SEM. * *p* < 0.05; ** *p* < 0.01 vs. the control group (5 mM) of 24 h. (ERK:N = 4).

## Data Availability

The data underlying this article will be disclosed to the corresponding author upon reasonable request.

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
