# Peer review of "Empagliflozin Prevent High-Glucose Stimulation Inducing Apoptosis and Mitochondria Fragmentation in H9C2 Cells through the Calcium-Dependent Activation Extracellular Signal-Regulated Kinase 1/2 Pathway"

_ijms, 2024, doi:10.3390/ijms25158235_

Round 1

Reviewer 1 Report

Comments and Suggestions for Authors

Review for the manuscript “Empagliflozin Prevent High-Glucose Stimulation Inducing Apoptosis and Mitochondria Fragmentation in H9C2 cells through the Calcium-Dependent Activation Extracellular Sig- nal-Regulated Kinase 1/2 Pathway ”

 The authors aimed to evaluate if empagliflozin could prevent high glucose induced mitochondria fragmentation through calcium-mediated activation of extracellular signal-regulated kinase 1/2 (ERK 20 1/2) in H9C2 cells. The authors found that Empagliflozin could reverse the apoptosis effect of HG stimulation on H9C2 cells. Also, mitochondria fragmentation  caused by HG condition was reduced by empagliflozin. When the authors investigated the expression of mitochondria fission protein and ERK 1/2 under HG stimulation, they found that the expression of these proteins were decreased under empagliflozin treatment.

Comments

The manuscript is written in a cursive manner, with many details about the laboratory methods used, so that it can be reproduced by other laboratories.

The authors present the results in the form of 6 clearly presented figures.

The results are useful to clinicians.

The conclusions support the results obtained.

Reviewer 2 Report

Comments and Suggestions for Authors

I read with interest the manuscript “Empagliflozin Prevent High-Glucose Stimulation Inducing 2 Apoptosis and Mitochondria Fragmentation in H9C2 cells 3 through the Calcium-Dependent Activation Extracellular Sig- 4 nal-Regulated Kinase 1/2 Pathway”.

Minor English revision is required.

1.       Discussion of Results: The findings are discussed in the context of their impact on mitochondrial dynamics, which is insightful. However, linking these results to broader physiological outcomes would strengthen the discussion.

2.       Mechanistic Insight: The discussion of calcium’s role in ERK1/2 activation is insightful. However, more detail on how empagliflozin modulates calcium signaling would be beneficial.

3.       Mechanistic Explanation: The results suggest a clear mechanistic pathway by which empagliflozin exerts its protective effects. However, further validation using additional molecular techniques (e.g., siRNA knockdown of ERK1/2) would strengthen these conclusions.

4.       Future Directions: Suggestions for future research, such as in vivo studies or exploration of other pathways, would be a valuable addition.

Comments on the Quality of English Language

minor

Round 2

Reviewer 2 Report

Comments and Suggestions for Authors

Satisfied.